# Dense Convolutional Neural Network for Identification of Raman Spectra

**DOI:** 10.3390/s23177433

**Published:** 2023-08-25

**Authors:** Wei Zhou, Ziheng Qian, Xinyuan Ni, Yujun Tang, Hanming Guo, Songlin Zhuang

**Affiliations:** Engineering Research Center of Optical Instrument and System, Ministry of Education, Shanghai Key Laboratory of Modern Optical System, School of Optical-Electrical and Computer Engineering, University of Shanghai for Science and Technology, 516 Jungong Rd., Shanghai 200093, China

**Keywords:** deep learning, Raman spectroscopy, dense network, interfered spectra

## Abstract

The rapid development of cloud computing and deep learning makes the intelligent modes of applications widespread in various fields. The identification of Raman spectra can be realized in the cloud, due to its powerful computing, abundant spectral databases and advanced algorithms. Thus, it can reduce the dependence on the performance of the terminal instruments. However, the complexity of the detection environment can cause great interferences, which might significantly decrease the identification accuracies of algorithms. In this paper, a deep learning algorithm based on the Dense network has been proposed to satisfy the realization of this vision. The proposed Dense convolutional neural network has a very deep structure of over 40 layers and plenty of parameters to adjust the weight of different wavebands. In the kernel Dense blocks part of the network, it has a feed-forward fashion of connection for each layer to every other layer. It can alleviate the gradient vanishing or explosion problems, strengthen feature propagations, encourage feature reuses and enhance training efficiency. The network’s special architecture mitigates noise interferences and ensures precise identification. The Dense network shows more accuracy and robustness compared to other CNN-based algorithms. We set up a database of 1600 Raman spectra consisting of 32 different types of liquid chemicals. They are detected using different postures as examples of interfered Raman spectra. In the 50 repeated training and testing sets, the Dense network can achieve a weighted accuracy of 99.99%. We have also tested the RRUFF database and the Dense network has a good performance. The proposed approach advances cloud-enabled Raman spectra identification, offering improved accuracy and adaptability for diverse identification tasks.

## 1. Introduction

The rapid developments and widespread applications of cloud computing and deep learning are promoting the transformation in all walks of life from traditional modes to intelligent modes. In the field of Raman spectroscopy, there is a widely concerned application mode. That is, transforming the traditional expensive terminal instruments, with complete software and hardware functions, into cheap ones with simple software and hardware. The cheap terminal instruments could transmit the collected data to the cloud servers through the cloud networks. The cloud servers would quickly identify the spectra due to their powerful computing abilities, abundant spectra databases and intelligent analysis algorithms. And then, the cloud servers would feed back the results to the terminal instruments. The realization of this vision urgently requires the development of deep learning methods applicable to the field of Raman spectroscopy.

For achieving fast spectral identification speeds and high accuracies, deep learning methods have found many applications in Raman spectroscopy identification. Notably, methods based on convolutional neural network show great advantages [1]. Since Liu et al. published the application of convolutional neural networks with a five-layer structure for Raman spectrum recognition in 2017 and demonstrated its superiority [2], many scholars have proposed a variety of CNN-based models in this area, applying them in different fields. Huang et al. constructed a five-layer model to classify species by analyzing the Raman spectra of blood [3], while Zhang et al. [4] introduced a transfer learning application also with a five-layer structure model to solve the problem of number limiting in a Raman spectra database. Fan et al. [5] proposed the DeepCID solutions for component identification in mixtures with a network of a seven-layer structure. Lyu et al. used a five-layer CNN model to identify seal authenticity and reached a 100% recognition rate [6]. Wu et al. established a five-layer network to identify the content of olive oil, providing a new analytical method for quantitative identification [7]. In the field of the Raman spectra identification of bacterial samples and tumor samples, there is abundant relevant literature proving the practicability and superiority of CNN-based models [8,9,10,11,12,13,14,15,16,17]. However, the proposed models in the previous literature described above [2,3,4,5,6,7,8,9,10,11,12,13,14,15,16,17] all have shallow networks with 12 layers and simple structures. On the premise of avoiding over-fitting and improving the generalization ability of the model, a deeper network can better extract the features of the targets to obtain better classification results [18]. Especially for the Raman spectra detected in complex environments with a simple terminal instrument, which are likely to be greatly interfered with and show a certain difference compared to the standard ones, the accuracy of identification by shallow network models would be significantly reduced.

To be more specific, the complexity of Raman spectra application scenarios might be found in the presence of various noises and the fact that spectrometer users may not be professionals. Factors like operation modes, vessels, packaging and detection postures would also bring unexpected interferences. When the noises and interferences increase, the prediction function should give less weight to those interfered wavebands of the spectra. A deeper network is needed to accurately carry out spectra identification under these complex conditions. In theory, while the depth and complexity of a network increases, its receptive field will continue to grow, and its feature extraction ability will continue to be enhanced [10,18,19]. The prediction function is expected to be better fitted as more parameters are introduced. After training, the most appropriate network would be found in order to resist the influence of interference and make a more accurate classification prediction.

In this paper, a deep learning algorithm based on a Dense convolutional neural network has been proposed. The network provides a consistent and efficient solution for achieving fast and accurate Raman spectra identification. The Dense network has a very deep construction of over 40 layers in three main function parts. It provides plenty of parameters to adjust the weight of different wavebands. It can minimize the influence of interfered wavebands and accurately focus on the real Raman peaks. The kernel Dense blocks part has four Dense blocks with a feed-forward fashion of connection for each layer to every other layer. Each block contains shorter connections between layers close to the input and layers close to the output. It can alleviate the gradient vanishing or explosion problems, strengthen feature propagations and encourage feature reuse. And it substantially reduces the number of parameters compared to the same depth networks to make the model easily trainable. The Dense network can rapidly and accurately identify the Raman spectra of normal samples. And it shows more accuracy and robustness compared to other CNN-based algorithms for interfered Raman spectra. We take samples of liquid chemicals as interfered examples and set a database of 1600 Raman spectra of 32 different kinds of liquid compound. They were detected in different detection postures. While those simple shallow networks generally perform according to the different interfered spectra caused by different detection postures and other factors of the same sample, the Dense network outputs stable and accurate identification results because it has a special deep complex network structure. In the actual test, the Dense network achieved 100% accurate spectral identification in 49 results of 50 repeated experiments, with only two misjudgments in one experiment, resulting in a weighted accuracy of 99.99%. The Dense network was also subjected to testing with the RRUFF database, demonstrating exceptional performance. This work is anticipated to effectively promote the application of deep learning in the domain of Raman spectroscopy, especially in substance identification such as tumor diagnosis, real and fake drug identification, chemical identification and so on.

## 2. Methods

### 2.1. Database

This study employs Raman spectra of liquid chemicals contained within vessels as an example to verify the Dense network’s ability in identifying interfered Raman spectra. The samples consisted of 32 different kinds of transparent liquid chemicals, including 1-dodecanethiol, 1-octadecene, 1,3,5-trimethylbenzene, *N*,*N*-dimethylformamide, ethyl butyrate, triethylamine, acetone, ethylene glycol, acetonitrile, butyl acetate, isoamyl acetate, acetate ethyl, isobutyl acetate, methylacetate, diethylamine, methyl sulfoxide, dimethylbenzene, acetic acid, pyrrolidine, pyridine, tetrahydrofuran, ethylisobutyrate, 3-methyl-1-butanol, butylamine, butyl alcohol, 1-propanol, chlorbenzene, 9-octadecenylamine, cyclohexane, formic acid, benzene and aniline. They were all put in common small cylindrical bottles. The material of the vessels will also have certain Raman spectra peaks, which will cause great interference to the Raman spectra of the detected samples. Furthermore, the different detection postures (bottle upright, side or bottom probe) will make the relative intensity of the sample’s Raman peaks and vessel interference change differently, leading to different signal-to-noise ratios. These factors would make the detected Raman spectra very complex and varied, and quite different from the standard ones. This condition is in accordance with the actual application scenarios of the cheap terminal instruments based on cloud computing.

All Raman spectroscopy data acquisition experiments were carried out using a portable Raman spectrometer (Model: FPRam8000) designed and produced by Wuhan Feipu Photoelectric Technology Co., Ltd. (Wuhan, China). This spectrometer uses the Hamamatsu S11510 CCD image sensor as the detector camera and has a fiber-optic Raman probe. The laser has a power range of 10 mW to 400 mW which is adjustable and the excitation wavelength is 785 nm. The spectral range is of 170–3200 cm^−1^ (Raman shift) and the resolution is about 7 wavenumbers. Every spectrum has a total of 2068 data points. The spectra are detected at room temperature and each sample was detected in three different postures: bottle upright, side or bottom probe (as shown in Figure 1).

According to the actual application scenarios, we cannot determine the specific situation of the spectrometer users and how the target samples are detected. The relative intensity of the bottle material’s interference and the sample’s Raman peaks will change due to the different specific detection scenes. That is, the detected spectra will be various. So, we set the database in different optical powers and integration times, as well as different detection postures. We detected the Raman spectra of samples as richly as possible. These operations would collect the spectra of different relative intensities between the interferences and sample’s Raman peaks. These operations are aiming to simulate the different specific situations in which the spectrometers are used and enriching the database would help to train a robust and stable model. The model would accurately identify the approximate spectra subsequently detected by the actual spectrometer users. Finally, we detected more than a dozen spectra for every sample in each posture, 50 spectra for the three postures in total. The Raman spectra of ethylene glycol is shown in Figure 2 as an example.

It can be seen from Figure 2 that the intensity of interference caused by the bottle material is lower when the laser is injected from the bottom of the bottle for Raman detection. The interference intensity of the bottle is stronger when the bottle is upright or on its side, some of which even exceeds the intensity of Raman peaks of the liquid chemical itself. That is, the Raman peak of the bottle material is higher than the primary Raman peak of the actual sample. These are all in accordance with realistic possibilities. It can also be intuitively observed from the Raman spectra that it is, indeed, a great challenge to accurately identify Raman spectra under such strong interference.

Ultimately, we collected a total of 1600 Raman spectra. They are all in the wavenumber range of 170–3200 cm^−1^ with an intensity of 0 to 1 by individual normalization. Parts of them have been shown in Figure 3. All the collected spectra comply with the detection scenario of practical application and algorithms, like shifting wavenumbers or replication to expand the spectra database, are not used.

### 2.2. Dense Network Model

A Raman spectrum is taken as an input *x* into the network and the model will predict an output *y* as the prediction class of this spectrum in the Raman spectral identification method. In this paper, we train a model *f* that predicts values *y*′ = *f*(*x*), where *y*′ is the estimate of ground truth output *y*.

As demonstrated in Figure 4, Dense networks consist of 3 main functional parts, which are named embedding, Dense blocks and exportation, respectively. We used *f*_1_, *f*_2_, *f*_3_ to denote each part and our model can be denoted as *f*(*x*) = *f*_3_(*f*_2_(*f*_1_(*x*))).

The first function, named embedding, takes the input spectrum *x* and represents it as a set of feature maps (*L*_3_) through three layers of convolution. It can be denoted as *L*_3_ = *f*_1_(*x*). The embedding part expands the number of channels and becomes prepared for subsequent computations. The convolution layer could be expressed as follows:(1)$$y=f\left(\sum_{i}w_{i}*x_{i}+b_{i}\right),$$
here, it convolutes *x* with weight *w* and then adds bias *b*. As shown in Figure 4, the parameters of the three convolution layers are, respectively, (16,5), (32,5) and (64,5). This means the kernel sizes of the weight *w* are all 5 and the out channels of each layer are, respectively, 16, 32 and 64. And the LeakyReLU function, which has been applied as an activation function in the embedding part with the parameter alpha as 0.3, is described as:(2)$$\mathrm{LeakyReLU}(x)=\left\{\begin{array}{l} x\;\;\;\;\;,\;x>0\\ \alpha x\;\;\;,\;x\leq 0 \end{array}\right.,$$

The MaxPooling method (defined in Equation (3)) has also been applied after each convolution layer with the pool_size of 2 and strides of 2. Pooling can be regarded as non-linear down-sampling, which can reduce the size of the representations, the number of parameters and the amount of computations [20].
(3)$$MaxPooling\left(x\right)=\max\left(x_{i,j},x_{i+1,j},x_{i,j+1},x_{i+1,j+1}\right)$$

After a batch-normalization operation, we come to the Dense blocks part. We have four Dense blocks in this part. It is the core part of the entire model. Four Dense blocks have the same structure as, but different parameters to, the convolution kernel. The parameters of every convolution stay fixed in the same block and double in the next block. The details of the Dense block are shown in Figure 5. It can be seen that there are 4 nodes in one block. The output layer from the previous step enters the block as input layer *L*. To keep the number of channels in the block consistent, we convoluted *L* with a kernel channel of *N* and size of 1. The result layer is node 1. On the other path, *L* has to be convoluted twice with kernel channel of *N* and size of 3 and obtain the result layer *L*_2′_. We add up node 1 and *L*_2′_ together to obtain node 2. This constitutes a residual structure. With three residual structures, we obtain four nodes. But, different from a residual network, we introduce direct connections from any layer to all subsequent layers. Consequently, the *l*th node receives the feature-maps of all preceding nodes as input. So, the Dense network can be denoted as follows:(4)$$x_{l}=Sum(x_{1},x_{2},\ldots ,x_{l-1}),$$

Node 3, for example, would be the sum of node 1, node 2 and the convolution result *L*_4′_. After node 4 has been batch-normalized, it becomes the output of this block. In the first block, the feature num we choose is 64. This means the channel of all the layers in this block is 64. And the model has 4 blocks in total and the feature number of the last block is 512. As shown in Figure 4, *L_out_* is the final output of the Dense blocks with the channels of 512 to be the input to the next part. *L_out_* has undergone *max_pooling* and can be denoted as *L_out_* = *f*_2_(*L*_3_).

The Flatten method turns the multidimensional nested array into a one-dimensional array. The convolution layer is transferred to the fully connected layer using the Flatten method. In the exportation part, there are two dense layers each with a unit of 1024 and one with ReLU activation. Then, the final identification prediction has been carried out using Softmax function, which is defined as follows:(5)$$SoftMax\left(x,n\right)=\frac{e^{w_{k}x}}{\sum_{i=1}^{n}e^{w_{i}x}},$$

The exportation part can be denoted as *y*′ = *f*_3_(*L_out_*). Now we have all components for the Dense network model. The detailed parameters are presented in Table 1.

### 2.3. Training

Loss function has been introduced to measure the degree of difference between the predicted results *y*′ of the samples’ classes and the true class values *y*. The target of training is to search for the most appropriate combination of parameters to minimize the loss function. Categorical cross-entropy [21] loss function has been applied in this article which is defined as follows:(6)$$L\mathrm{oss}=-\sum_{i}y_{i}\cdot \log\left(y\prime_{i}\right),$$

In principle, the loss function decreases and gets closer to zero while the model is trained to perform better. The adaptive moment estimation (Adam) [22] is used as an optimizer since it requires little memory and has high computational efficiency. The learning rate of the Adam optimizer is set to 0.0001.

The 1600 spectra are split randomly into a training set and test set in the fixed ratio of 7:3 according to category labels. The split is based on both the liquid sample and position. It can ensure that each category has samples in the training set and test set around the approximate proportion. To evaluate the fit of the model being trained and understand the situation of model training, 20 percent of the data in the training set are further split as the validation set. As a result, in general, a ratio of training set, validation set and test set is selected at 56:14:30. The batch size is set to 32 and the weights and biases are initialized randomly. Early stopping is applied to prevent overfitting and save time. The upper limit epoch is set at 200. If the prediction accuracy of the validation set does not improve within 20 epochs, the training would stop. The loss, recall, precision and AUC epoch curve are shown in Figure 6. The model has converged rapidity within 10 epochs.

Because the Dense network connects each layer to every other layer in a feed-forward fashion in the Dense blocks, it can alleviate the gradient vanishing or explosion problems, strengthen feature propagations, encourage feature reuses and substantially reduce the number of parameters compared to same depth networks. And these make the model easily trainable and show great robustness.

The Dense network is implemented based on NumPy and Keras in Python 3.8. The training is performed on the operating system Ubuntu 16.0 with a single NVidia GTX Titan GPU, Tan Intel Core i9-7900X processor and 256 G DDR4 memory.

## 3. Results and Discussion

Due to the random initialization of the model parameters and the random splitting of the dataset, there would be some differences in the identification results from each time of model training and prediction. Therefore, we have conducted 50 repeated training and testing sets without repetition. In 49 of these experiments, the Dense network can achieve 100% accurate spectral identification results, and only in 1 experiment were two misjudgments were generated, resulting in a weighted accuracy of 99.99%. After training, the Dense network can identify hundreds of spectra within one minute. The prediction confusion matrix is shown in Figure 7. It can be seen in the figure that the squares have a diagonally linear distribution. This means that there is a high degree of consistency between the predicted classes and the actual ones of these chemical spectra. The result proves the reliability and veracity of the Dense network.

For the common CNN-based models of Raman spectra identification, the algorithm methods mainly find the primary peak of the Raman spectrum and give it a higher computational weight. In the process of model training, the convergence can be carried out quickly through the primary peak matching. Then, the models could recognize and identify the Raman spectra. However, for the Raman spectra which are greatly interfered by the bottle material, it is easy to misjudge the primary peak in these models. The intensity of the bottle material would probably be higher than the primary peak in the spectra. This would easily lead to gradient vanishing or explosion in training. Those simple shallow structures of networks like Liu’s [2], Zhang’s [4] and Fan’s [5] would not be suitable for this kind of situation.

We run all the models the same way 50 times and the identification result is shown in Table 2. In an experiment mentioned in Table 2, the model would be trained and then would identify the test dataset. When the identification accuracy is less than 20%, the model would not have completed the training efficiently. This experiment will be counted as an ‘error’, meaning the model could not effectively output the spectra identification results. Under this condition, the model’s gradient would probably vanish or explode. The ‘error rate’ revised in Table 2 is defined as the ratio of the ‘error’ times to the total number of experiments (50).

The Dense network is robust. The training was effectively completed in every experiment. Therefore, the error rate of the Dense network is zero.

The main reason why the results of the other compared models are not ideal is that their models have sometimes not effectively converged. This may be because those structures are too shallow and the parameters are insufficient, which easily results in gradient explosion or vanishing in the training process, so that the fitting convergence cannot be completed. But the Dense network has a very deep structure of over 40 layers. It provides plenty of parameters to train the model and obtain accurate identification results. The Dense blocks have feed-forward fashions of connection for each layer to every other layer. Each block contains shorter connections between layers close to the input and layers close to the output. It can alleviate the gradient vanishing or explosion problems, strengthen feature propagations and encourage feature reuse. In the process of training, it has low sensitivity to the interference waveband and assigns appropriate weights to different wavebands. The Dense network shows more accuracy and robustness (fewer errors) compared to other CNN-based algorithms.

We supplemented the experimental results based on the RRUFF spectral database (which is consistent with the database used in the compared literature [2,4]) and made a comparison. We used the same selection method as Sang et al. [18]. This database contains 1332 classes of 8578 Raman spectra from well-characterized minerals.

The loss, recall, precision and AUC curves during model training are shown in Figure 8 and the compared result is shown in Table 3. The accuracy of the Dense network has a slight lead over the compared models.

The Dense network could be applied to other kinds of Raman spectra identification. For other spectral data, the Dense network is also considered to be able to obtain a high identification accuracy after training. For the interfered spectra, as we detected in the Section 2.1, the high robustness of the Dense network makes its accuracy state-of-the-art.

## 4. Conclusions

In this study, the Dense network has been proposed to identify Raman spectra with environment noises and interferences based on the application mode of cloud computing with cheap terminal instruments and realistic complex detection scenarios. The proposed method has a very deep network structure of over 40 layers in order to reduce the various interferences and it shows great identification accuracy and robustness. The Dense network introduces direct connections between any two layers with the same feature map size in each Dense block. The Dense network can capture high-frequency Raman peak signals, with low sensitivity to interference, due to its deep and special structure. The model is easily trainable. It has a weighted identification accuracy of 99.99% in the 50 experiments based on the Raman spectra database of interfered liquid chemical samples. The Dense network also has certain advantages tested on the RRUFF database. It can be affirmed that the Dense network would provide an effective and consistent way to identify interfered Raman spectra from complex detection environments. In the future, further experiments and optimizations should be conducted. The Dense network would be able to accurately identify various spectra of different interferences in complex environments of different samples, including the spectra of mixtures and hydro-solutions. We believe that the Dense network can be applied to a wider range of real-world scenarios, and effectively provide a solution for terminal instrument convenience and cloud computing applications. The Dense network can also be applied in other fields of Raman spectra identification.

## Figures and Tables

**Figure 1 sensors-23-07433-f001:**
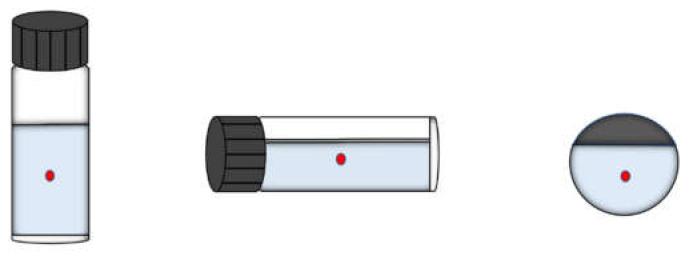
Three different detection posture (bottle upright, side or bottom probe). The red point refers to the incident excitation laser.

**Figure 2 sensors-23-07433-f002:**
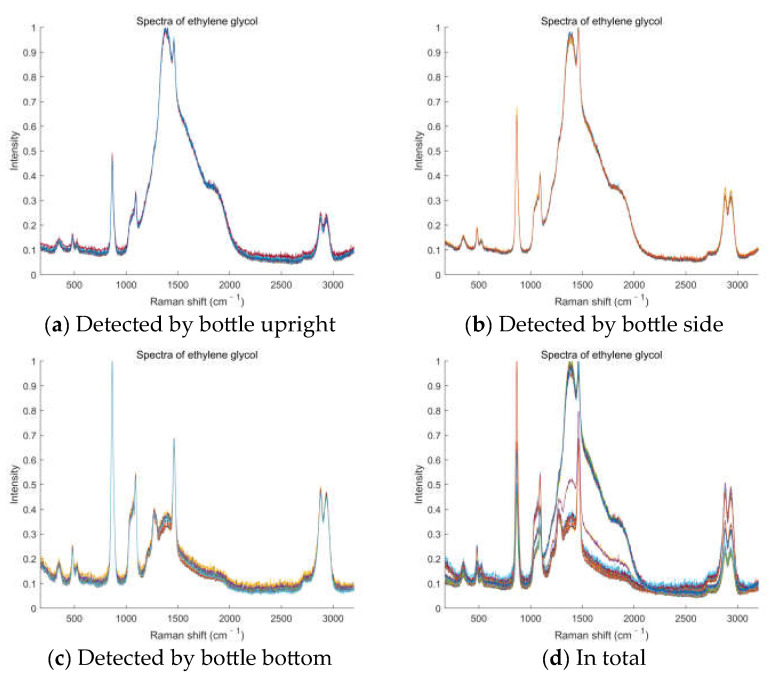
The Raman spectra of ethylene glycol in different detection postures and in total. The different color lines in the figure represent different spectra. From the wavenumber of 1000 to 2000, the Raman curves are all strongly interfered with by bottle materials.

**Figure 3 sensors-23-07433-f003:**
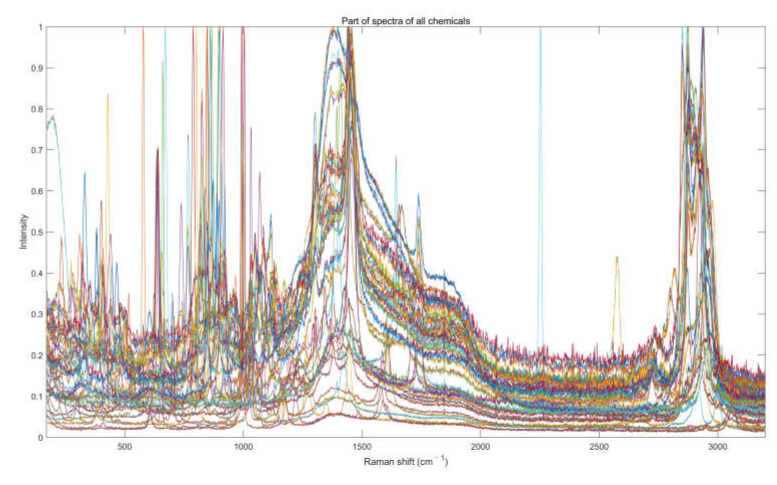
Parts of all Raman spectra. The different color lines in the figure represent different spectra.

**Figure 4 sensors-23-07433-f004:**
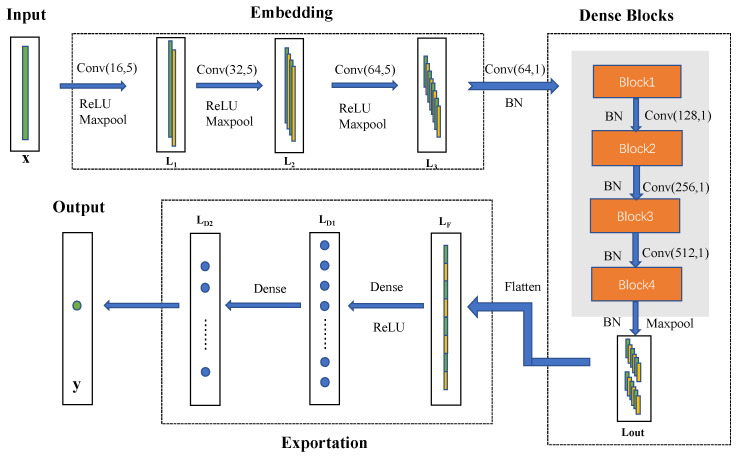
The structure of the Dense network.

**Figure 5 sensors-23-07433-f005:**
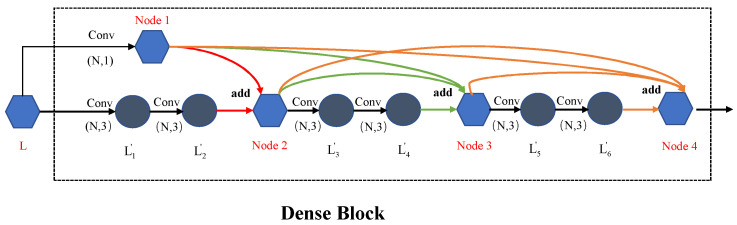
The details of the Dense blocks.

**Figure 6 sensors-23-07433-f006:**
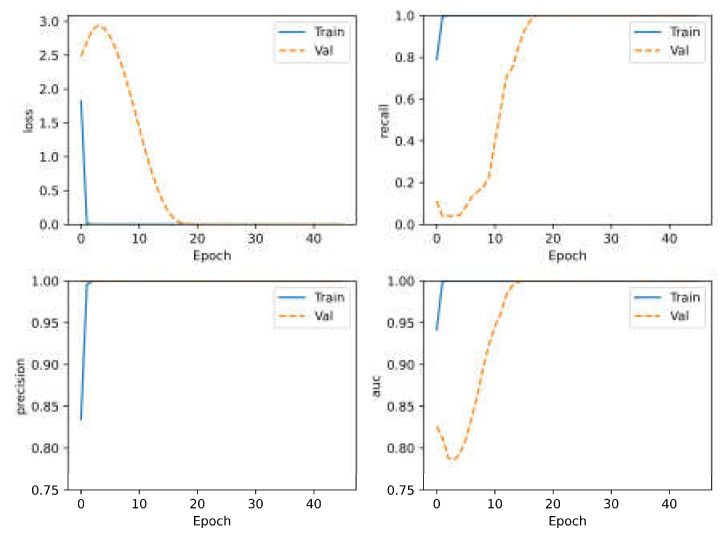
The loss, recall, precision and AUC curves during model training. As the number of training sets increases, the four curves all rise (or fall) quickly and soon stabilize.

**Figure 7 sensors-23-07433-f007:**
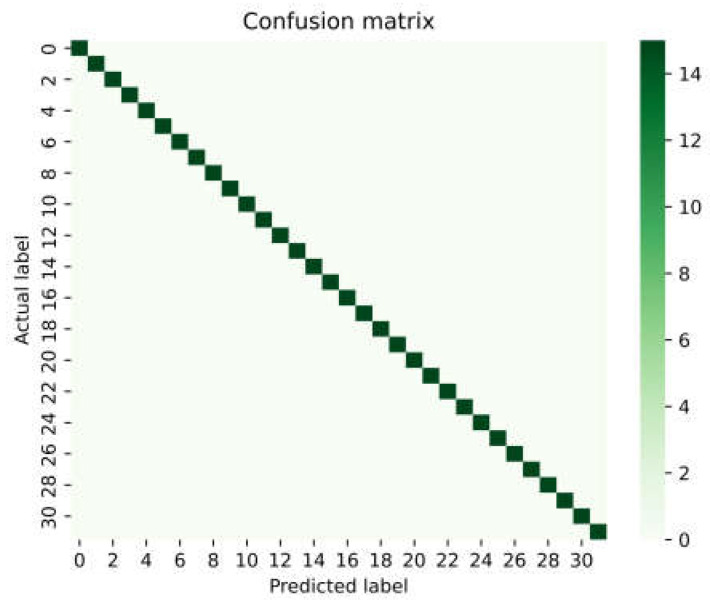
The confusion matrix of prediction results.

**Figure 8 sensors-23-07433-f008:**
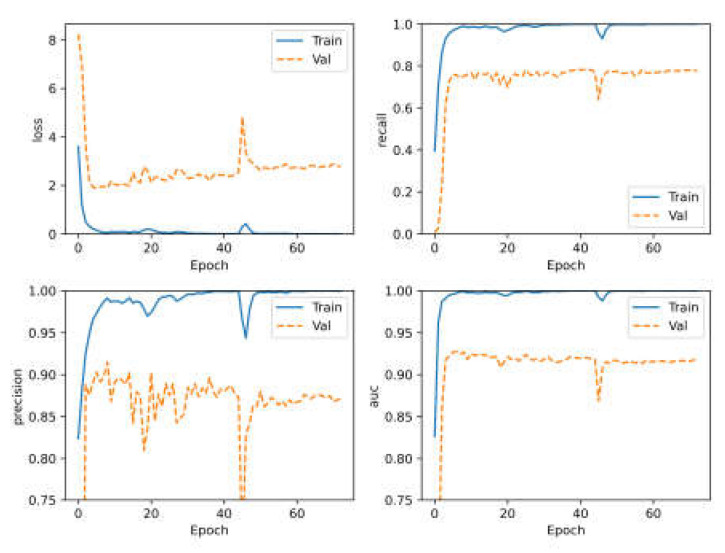
The loss, recall, precision and AUC curves during model training on database of RRUFF.

**Table 1 sensors-23-07433-t001:** Detailed information about the parameters of the Dense network.

Part	Layers	Type	Output Shape	Parameters
	Input	Input	(Batch, 2068, 1)	-
Embedding	L_1_	Conv1D	(Batch, 2068, 16)	(16, 5, 1)
		Max_pooling1d	(Batch, 2068, 16)	(2, 2)
	L_2_	Conv1D	(Batch, 1034, 32)	(32, 5, 1)
		Max_pooling1d	(Batch, 1034, 32)	(2, 2)
	L_3_	Conv1D	(Batch, 517, 64)	(64, 5, 1)
		Max_pooling1d	(Batch, 517, 64)	(2, 2)
Dense blocks	Block 1	Conv1D	(Batch, 258, 64)	(64, 3, 1)
		Normalization	(Batch, 258, 64)	256
	Block 2	Conv1D	(Batch, 258, 128)	(128, 3, 1)
		Normalization	(Batch, 258, 128)	512
	Block 3	Conv1D	(Batch, 258, 256)	(256, 3, 1)
		Normalization	(Batch, 258, 256)	1024
	Block 4	Conv1D	(Batch, 258, 512)	(512, 3, 1)
		Normalization	(Batch, 258, 512)	2048
		Max_pooling1d	(Batch, 129, 512)	(2,2)
Exportation	L_out_	Flatten	(Batch, 66,048)	-
	L_d1_	Dense	(Batch, 1024)	(1024)
	L_d2_	Dense	(Batch, 1024)	(1024)
	Output	Dense	(Batch, #classes)	(#classes)

**Table 2 sensors-23-07433-t002:** The results of comparation between the Dense network and other models on liquid chemicals in 50 experiments.

	**Liu et al. [2]**	**Fan et al. [5]**	**Zhang et al. [4]**	**Dense**
accuracy	0.7925	0.8383	0.8260	0.9999
error rate	0.22	0.34	0.40	0

**Table 3 sensors-23-07433-t003:** The results of comparison between the Dense network and other models on RRUFF database.

	**Liu et al. [2]**	**Fan et al. [5]**	**Zhang et al. [4]**	**Dense**
accuracy	0.8309	0.8368	0.8466	0.8601

## Data Availability

The data that support the findings of this study are available from the authors on reasonable request (see Author Contributions for specific data sets).

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
