# Peer review of "Dense Convolutional Neural Network for Identification of Raman Spectra"

_sensors, 2023, doi:10.3390/s23177433_

Round 1

Reviewer 1 Report

The manuscript presents a computational method for Raman spectra identification. The conclusion might be relevant, however it would be more appropriate to test the presented method on naive datasets. As the manuscript states, the accuracy of the presented method is remarkable in the sense that it can identify spectra from the same cluster that was used for training. What about spectra of different provenience (even different lab or system)? The spectra might be contaminated with background from the material holder or container. How these backgrounds were treated by the model? Were any pre-background removal applied? The used method is compared to literature data, however running the literature methods on the same spectra dataset would be more suitable to prove the manuscript's conclusions. 

English language is fine of the manuscript, however in some cases there are few incomplete sentences. Some minor spell check and quality check by native speaker would be benefic.

Reviewer 2 Report

Although the authors seem to be very efficient with Neural Networks, I am sorry to say that I am unable to place my trust in a model that exhibits zero error, achieves convergence within a mere 10 epochs, and possesses a faultless confusion matrix as depicted in Figure 7, particularly in light of the observed diversity illustrated in Figure 3.

General issues:

The code is not shared.

Where might one use the results of this study? The substances in the samples are pure, unlike the more common practice of using mixtures or hydro-solutions. How well does the code run under those conditions? Is there room for growth in that respect? 

Were the train and test sets split solely based on the liquid sample? Is there any division or separation resulting from the measurement that is based on position? The bottle material appears to have a significant impact on the result. What would be the weight of this factor in the final outcome? It seems that it has been normalized or neglected!

Specific comments:

L105: please provide a table of all the chemicals used along with a representative spectrum (possibly as supplementary information). The reader must be able to explore differences or similarities between spectra.

L128-133: The statements in this paragraph are incorrect. Furthermore, I cannot understand the rationale behind the intensity of the main peak. Does the Dense Convolutional Neural Network (CNN) focus on detecting the primary peak of the compound? If that is the case, what is the motivation behind “to obtain Raman spectra of the same sample as rich as possible” (L137)? In Figure 2, which peak is identified as the "primary Raman peak"? If there is an overlap observed within the spectra of distinct compounds, it raises the question of how to effectively discern and differentiate between the respective substances. On the other hand, if the compounds have clearly different spectra, then the identification might be biased. Overall, the whole paragraph in question exhibits an alarming degree of confusion. 

Table 2: I wonder how feasible a zero error can be. Please provide the code with a minimal dataset so that the execution efficiency can at least be evaluated. Furthermore, I searched for Ref5 which gives ~99% accuracy in their paper but only ~82% in your samples. What would be an explanation for that discrepancy?

Please use plain English instead of hyper-complicated sentences.

Round 2

Reviewer 2 Report

I would like to thank the authors for the detailed reply. In my opinion the manuscript has been improved and hopefully the authors share my view.

The only comment I have is in L107-108 where all the chemicals should be written. The spectra which the authors kindly included in a zip file can be attached as supplementary information. 

 Minor editing of English language is required
